# Effect of Processing on Residual Buprofezin Levels in Ginseng Products

**DOI:** 10.3390/ijerph18020471

**Published:** 2021-01-08

**Authors:** Hyun Ho Noh, Hyeon Woo Shin, Dong Ju Kim, Jeong Woo Lee, Seung Hyeon Jo, Danbi Kim, Kee Sung Kyung

**Affiliations:** 1Chemical Safety Division, Department of Agro-Food Safety and Crop Protection, National Institute of Agricultural Sciences, Rural Development Administration, Wanju 55365, Korea; Noh1983@korea.kr (H.H.N.); danbi6334@korea.kr (D.K.); 2Department of Environmental and Biological Chemistry, College of Agriculture, Life and Environment Science, Chungbuk National University, Cheongju 28644, Korea; gusdn950725@naver.com (H.W.S.); kimdj6746@naver.com (D.J.K.); leejung2648@naver.com (J.W.L.); jj3348@naver.com (S.H.J.)

**Keywords:** buprofezin, ginseng, preharvest interval, drying processing, pesticide residue, processing factor, reduction factor

## Abstract

This study determined residual buprofezin levels in fresh ginseng and evaluated their changes during processing. Supervised field trials were conducted at Yeongju, Geumsan, and Goesan, Korea. Buprofezin 12.5% EC was applied to 5-y ginseng in accordance with the Korean good agriculture practice (GAP). Samples were collected at 0, 7, 14, 21, and 30 d after the final application. On day 14 (GAP-equivalent preharvest date), the ginseng was processed to obtain dried and red ginseng. The average buprofezin concentrations on day 0 were 0.076 (Yeongju), 0.055 (Geumsan), and 0.078 mg kg^−1^ (Goesan). Residual concentrations increased as ginseng was processed into dried and red ginseng. Residue levels in dried ginseng manufactured by hot air drying were higher than in red ginseng obtained by steaming, hot air, and sunlight drying. However, the absolute amount of pesticides decreased by approximately 20–30% as a result of calculating the reduction factor considering the dry yield and moisture content. Therefore, the residual concentration in processed products may vary depending on the processing method, and it is deemed necessary to consider the processing yield and moisture content when evaluating the safety of residual pesticides in dried processed products.

## 1. Introduction

Ginseng is cultivated in the United States, China, Japan, Russia, and Korea. Korean ginseng (*Panax ginseng C.A.* Meyer) is cultivated mainly in Korea, while *Panax quinquefolium* is grown mainly in Wisconsin and Minnesota, and *Panax japonicum* is grown mainly in Japan [1]. The major constituents of ginseng are saponins (3–6%), vitamins (0.05%), and carbohydrates (60–70%). Ginseng is administered for intestinal disorders and is an important traditional herbal medicine. It has pharmacological efficacy against fatigue and stress and exhibits immunity-enhancing and anti-cancer properties [1,2].

About 30% of all agricultural products worldwide are produced using pesticides. Crops must be protected from pests so that high yield and quality may be consistently assured [3,4]. When crops are grown without pesticides, fruit, vegetable, and cereal losses owing to pest infestations may be as high as 78%, 54%, and 32%, respectively [5]. Approximately 80% of the fruits and vegetables produced in the United States are treated with fungicides. The economic benefit of using fungicides is valued at $12,300 [6]. It has also been reported that if pesticides were not used in cotton, wheat, and soybean, exports would be reduced by 27% [3].

Many agricultural products go through processes such as washing, blanching, and stir-frying. The basic washing process is considered primary processing while blanching and stir-frying are secondary processing. Ginseng products are produced using the drying method and can also go through an additional extraction process, making it a tertiary processed product. As the amount of residual pesticides can decrease or increase during processing, various studies have focused on the residual pesticide concentration in processed agricultural products [7,8,9,10,11,12]. Moreover, the Codex, EU, and other national authorities, including US Environmental Protection Agency (EPA), consider the various processing steps while assessing pesticide risk and defining maximum residue limits in food products.

Ginseng is particularly susceptible to various pests and diseases as it is grown in the same place under dark, humid conditions for 4–6 y. Therefore, pesticide application is essential for ginseng production and quality assurance [13]. Ginseng is grown under artificial shade for several years. Hence, a relatively lower amount of the pesticides used is lost by UV overexposure and rainfall compared to other field crops. Therefore, systemically sprayed pesticides such as buprofezin and azoxystrobin may continuously accumulate in ginseng [7]. Moreover, pesticides applied to the soil before planting can be transferred to ginseng [13].

Ginseng is widely distributed as fresh ginseng but is often processed and supplied in the form of dried and red ginseng to facilitate storage and enhance its medicinal properties [8]. In Korea, ginseng is processed into dried and red ginseng and exported. As a result of the recent free trade agreement (FTA), the international commerce of agricultural products has increased. Therefore, it is necessary to ensure the safety of exported ginseng products.

Buprofezin (2-*tert*-butylimino-3-isopropyl-5-phenyl-1,3,5-thyadiazinan-4-one) is a thiadiazin insecticide that inhibits molting. It is widely used in Korea to control Comstock mealybug infestation (*Pseudococcus comstocki* (Kuwana)) in ginseng. It is sprayed in a form that is stable under both acidic and alkaline conditions and is also thermostable and photostable [14]. Hence, its residues may be very persistent in the crops and the soil. In addition, the concentration of residual pesticides such as buprofezin increases with decreasing moisture content.

So far, most processing studies have only calculated the processing factor, representing the ratio of residual concentration before and after processing. Even if the concentration of residual pesticides in processed foods decrease, it is not known whether the concentration of residual pesticides among processed foods decreased due to the processing yield or whether the absolute quantity decreased due to degradation and/or loss of pesticide. Moreover, ginseng is processed using different processes, which can lead to different pesticide residue patterns. Therefore, the aim of this study was to investigate the relative changes in buprofezin residue content in processed ginseng by measuring residual levels after spraying the insecticide onto 5-y ginseng according to the Korean preharvest interval (PHI).

## 2. Materials and Methods

### 2.1. Field Trials

The test pesticide was the insecticide buprofezin 12.5% EC, which is widely used to control certain insect pests in various crops grown in the Republic of Korea. The test crop was 5-y ginseng. The sites Yeongju (36°54′13.0″ N 128°34′39.1″ E), Geumsan (36°06′13.3″ N 127°32′07.8″ E), and Goesan (36°50′00.1″ N 127°38′33.5″ E) were selected for the field trial. There were major climatic and cultivation differences among sites. The linear distances between the sites were 75–130 km. The control and treatment plot dimensions were 48.6 (W = 1.8 m × L = 27 m) and 48.6 m^2^ (W = 1.8 m × L = 9.0 m × 3 replications), respectively. Each treatment plot comprised three unit plots. The unit plots were separated by 1.8 m-long buffer zones to prevent cross-contamination. A decline study was conducted in the same way. Based on the Korean GAP for buprofezin, the general ginseng harvesting time was set to 14 d after the final application. Details of buprofezin application on the ginseng plants are summarized in Table 1. Samples were collected 5× at 7-d intervals from the final treatment day. Fourteen days after the final treatment on the preharvest day, samples were also collected to be processed into dried and red ginseng.

### 2.2. Processing Procedures and Sample Preparation

Freshly harvested ginseng was washed with tap water to remove soil particles from the roots. To obtain dried ginseng, the fresh ginseng was washed and desiccated in a hot air drying machine at 60 °C to a moisture content <14%. To obtain red ginseng, fresh ginseng was steamed for 3 h at 98 °C, then desiccated in a hot air drying machine at 65 °C to a moisture content of ~50–55%. Smaller roots were removed from the dried samples and were further sun-dried to reduce the moisture content to <14%. To prepare analytical subsamples, fresh ginseng was mixed with dry ice in a blender. The dried and red ginseng samples were directly blended without dry ice. The subsamples were packed in self-sealing plastic bags and stored at −20 °C until subsequent analysis.

### 2.3. Reagents and Materials

Analytical grade buprofezin (99.00% purity, Cas No. 69327-76-0) was procured from Dr. Ehrenstofer GmbH (Augsburg, Germany). Acetonitrile (HPLC grade) for the standard dilutions, sample extraction, and the LC mobile phase for instrumental analysis, and HPLC grade formic acid (>98%) for the LC mobile phase were purchased from Merck (Darmstadt, Germany). QuEChERS extraction packets for sample extraction and d-SPE tubes for sample purification were purchased from Agilent Technologies (Santa Clara, CA, USA). Combi-514R (Hanil Scientific Inc., Gimpo, Korea) and Combi-408 (Hanil Scientific Inc., Gimpo Korea) centrifuges were used for 50- and 2-mL centrifuge tubes, respectively. The 2010 Geno/Grinder^®^ automated homogenizer (SPEX SamplePrep LLC, Metuchen, NJ, USA) was used for sample extraction. The 13-mm PTFE syringe filters were obtained from Advantec (Tokyo, Japan).

### 2.4. Extraction and Purification

A 10-g sample of fresh ginseng was weighed into a 50-mL conical Falcon^®^ tube (Corning Inc., Corning, NY, USA). Ten milliliters of acetonitrile were added, and the tube was capped and shaken at 1300 rpm for 5 min. Then, 4 MgSO_4_, 1 NaCl, 1 sodium citrate, and 0.5 g sodium hydrogen citrate sesquihydrate were sequentially added to the tube. The tube was then shaken at 1300 rpm for 1 min and centrifuged at 3000 rpm for 5 min. Ten grams of ground dried ginseng and ten grams of ground red ginseng were weighed into a 50-mL conical tube and soaked in 10 mL distilled water for 1 h. Then, 10 mL acetonitrile were added, and the tube was capped and shaken at 1300 rpm for 5 min. Then, 4 MgSO_4_, 1 NaCl, 1 sodium citrate, and 0.5 g sodium hydrogen citrate sesquihydrate were sequentially added to the tube. The tube was shaken again at 1300 rpm for 1 min and centrifuged at 4000 rpm for 5 min. To purify the extraction solution, a 1-mL supernatant aliquot was transferred to a 2-mL centrifuge tube. Then, 150 MgSO_4_, 50 end-capped C18, and 50 mg combined primary and secondary amines were added to the tube. The mixture was vortexed for 1 min and centrifuged at 13,000 rpm for 5 min. Then, 500 μL supernatant was mixed with 500 μL acetonitrile to prepare the matrix-matched sample. The mixed solution was passed through a syringe filter and subjected to LC-MS/MS analysis.

### 2.5. Optimization of Instrumental Analysis

As buprofezin has three nitrogen atoms, it could be analyzed by gas chromatography with a nitrogen-phosphorus detector (GC-NPD). However, in this study, LC-MS/MS was used for rapid and efficient analysis. A reverse-phase octadecyl silica column (L = 100 mm, 2.1 mm I.D., 2.7 µm particle size) was used for high peak resolution. The mobile phase was a 10:90 (*v*/*v*) mixture of distilled water and 0.1% (*v*/*v*) formic acid. Acetonitrile plus 0.1% (*v*/*v*) formic acid served as a protonation enhancer. A 10-µg kg^−1^ matrix-dependent standard solution was injected in infusion mode for scan analysis to optimize the multiple reaction monitoring (MRM) conditions for buprofezin. Two of the highest-intensity ion fragments were determined as quantitation and confirmation ions, respectively. The buprofezin precursor ion was observed at *m*/*z* 306.2, and the quantitation and confirmation ions were observed at *m*/*z* 116.1 and 201.1, respectively. Details of the analytical conditions are shown in Table 2.

### 2.6. Matrix Effect

Previous studies attempted to reduce the matrix effect (ME%) between samples and analytes caused by ionization in the MS detectors. The simplest and most effective method of offsetting ME is matrix-matched calibration [15].

Here, calibration curves were plotted according to the peak area obtained from the analysis of pure and matrix-matched standards by LC-MS/MS. The ME was calculated using the calibration curve slopes, which ranged from 1 to 100 μg L^−1^ at 7 points (Equation (1)) [16]. The ME expresses the degree of ion suppression or enhancement strength and may be low (–20% < ME% < +20%), medium (–50% < ME% < −20% or +20% > ME% > +50%), or high (ME% < −50% or ME% > +50%) [16].
(1)$$\mathrm{ME}\;\left(\%\right)=\frac{Smmsc\;-\;Spsc}{Spsc}\times \;100\;$$
where ME is the matrix effect; *Smmsc*, the slope of matrix-matched standard calibration; and *Spsc*, the slope of pure standard calibration.

### 2.7. Limits of Quantitation and Recovery

The limit of quantitation (LOQ) of the residual buprofezin analytical method was optimized at the lowest concentration of 5 µg kg^−1^. It had sufficient reproducibility and a signal-to-noise ratio >10. The recovery test was conducted to validate the method at 5 (LOQ), 50 (10 × LOQ), and 70 µg kg^−1^. The latter is the maximum residue limit (MRL) for Korean ginseng.

### 2.8. Processing and Reduction Factors

The processing factor indicates the change in residual pesticide concentration during processing. It is the ratio of the residual amounts of raw materials before and after processing. The processing factor is calculated using Equation (2) [17]. The reduction factor is expressed as a change in the absolute amount of pesticides remaining in the raw agricultural products during processing. The reduction factor is calculated as the residual concentration based on dry weight by considering the net yield and the water content using Equations (3)–(6).
(2)$$\mathrm{Processing}\;\mathrm{factor}=\frac{\mathrm{Residue}\;\mathrm{in}\;\mathrm{raw}\;\mathrm{product}\;\left(\mathrm{mg}/\mathrm{kg}\right)}{\mathrm{Residue}\;\mathrm{in}\;\mathrm{processing}\;\mathrm{product}\;\left(\mathrm{mg}/\mathrm{kg}\right)}$$
(3)$$\begin{array}{c} \mathrm{Residue}\;\mathrm{in}\;\mathrm{raw}\;\mathrm{product}\;\mathrm{based}\;\mathrm{on}\;\mathrm{dry}\;\mathrm{weight}\;\left(\mathrm{mg}/\mathrm{kg}\right)\\ =\frac{100\times \mathrm{residue}\;\mathrm{in}\;\mathrm{raw}\;\mathrm{product}\;\left(\mathrm{mg}/\mathrm{kg}\right)}{100-\mathrm{water}\;\mathrm{content}\;\mathrm{of}\;\mathrm{raw}\;\mathrm{product}\;\left(\%\right)} \end{array}$$
(4)$$\mathrm{Yield}\;\mathrm{of}\;\mathrm{manufacture}\;\left(\%\right)=\frac{\mathrm{Weight}\;\mathrm{of}\;\mathrm{processing}\;\mathrm{product}\;\left(g\right)\;\times 100\;}{\mathrm{Weight}\;\mathrm{of}\;\mathrm{raw}\;\mathrm{product}\;\left(g\right)}$$
(5)$$\begin{array}{l} \mathrm{Residue}\;\mathrm{in}\;\mathrm{processing}\;\mathrm{product}\;\mathrm{based}\;\mathrm{on}\;\mathrm{dry}\;\mathrm{weight}\;\left(\mathrm{mg}/\mathrm{kg}\right)\\ =\frac{\mathrm{Residue}\;\mathrm{in}\;\mathrm{processing}\;\mathrm{product}\;\left(\mathrm{mg}/\mathrm{kg}\right)\;\times \;\left(100-\mathrm{yield}\;\mathrm{of}\;\mathrm{processing}\;\mathrm{product}\left(\%\right)\right)}{100-\mathrm{water}\;\mathrm{content}\;\mathrm{of}\;\mathrm{processing}\;\mathrm{product}\;\left(\%\right)} \end{array}$$
(6)$$\begin{array}{l} \mathrm{Reduction}\;\mathrm{factor}\\ =\frac{\mathrm{Residue}\;\mathrm{in}\;\mathrm{processing}\;\mathrm{product}\;\mathrm{based}\;\mathrm{on}\;\mathrm{dry}\;\mathrm{weight}\;\left(\mathrm{mg}/\mathrm{kg}\right)}{\mathrm{Residue}\;\mathrm{in}\;\mathrm{raw}\;\mathrm{product}\;\mathrm{based}\;\mathrm{on}\;\mathrm{dry}\;\mathrm{weight}\;\left(\mathrm{mg}/\mathrm{kg}\right)} \end{array}$$

### 2.9. Statistical Analysis

One-way analysis of variance (ANOVA) of the pesticide residues was performed in SPSS v. 23 (IBM Corp., Armonk, NY, USA). Duncan’s multiple range test was performed at the *p* < 0.05 level to identify significant differences between treatment means.

## 3. Results and Discussion

### 3.1. Calibration Curve and ME

The calibration curve equation for the buprofezin standard dependent on the fresh ginseng extract solution, which ranged from 0.001 to 0.1 mg kg^−1^, was y = 75,780.2223x + 37.3711 (r^2^ = 0.9988). The calibration curve formulas for buprofezin in dried and red ginseng samples were y = 70,343.3323x + 38.5042 (r^2^ = 0.9996) and y = 72,132.0527x + 53.3565 (r^2^ = 0.9995), respectively. The calibration curve formula for the pure standard analyzed to calculate ME was y = 80,014.6367x + 36.2676. The ME was −5.2%, −12.1%, and −9.9% for fresh ginseng, dried ginseng, and red ginseng, respectively. Buprofezin ionization was found to be suppressed by the detector in all cases. All samples presented a low ME (Figure 1). Nevertheless, a matrix-matched standard was used for accurate quantification.

### 3.2. Recovery

The average buprofezin recovery from fresh ginseng was 89.5–105.9%, and the coefficient of variation (CV) was <3% (Table 3). The average buprofezin recovery from dried and red ginseng ranged from 83.9–91.0% and from 90.7–97.6%, respectively. The Food and Agriculture Organization of the United Nations (FAO) (2016) [18] set the effective recovery range and CV according to the fortification levels. If the fortification level was ≤0.001 mg kg^−1^, a recovery range of 50–120% and a CV <35% were valid. If the fortification levels were >0.001 or ≤0.01 mg kg^−1^, a recovery range of 60–120% and a CV <30% were recommended. A fortification level of 0.01 or ≤0.1 mg kg^−1^ required a recovery range of 70–120% and a CV of <20%. If the fortification level was >0.1 or ≤1.0 and >1.0 mg kg^−1^, a recovery range of 70–110% was required. Valid CVs were <15% and <10%, respectively. Therefore, the analytical method established to analyze buprofezin in fresh ginseng and its processed products (dried and red ginseng) was suitable because all recovery test results met these conditions.

### 3.3. Residues in Fresh Ginseng

Table 4 shows that the average residual buprofezin levels in fresh ginseng in Yeongju (Site 1) and Goesan (Site 3) were 0.076 ± 0.005 and 0.078 ± 0.002 mg kg^−1^, respectively, on the final spraying day. By day 7, the average residual buprofezin levels were only 0.018 ± 0.001 and 0.012 ± 0.000 mg kg^−1^, respectively. After that time, the average residual buprofezin levels were <LOQ until day 30. Experiments at Site 2 (Geumsan) showed that the average buprofezin levels in the ginseng were 0.163 ± 0.006 on the final day of spraying and 0.055 ± 0.002 mg kg^−1^ on day 14. The latter was the harvest day, according to the Korean preharvest interval (PHI). Thereafter, the residual buprofezin level continued to decrease and was only 0.014 mg kg^−1^ by day 30 after the final spraying. Representative chromatograms are presented in Figure 2.

The sprayed pesticides leave residues in and on the crops by various mechanisms. The most common scenario is direct crop exposure to the pesticide. Here, residual pesticide levels were measured in ginseng root after foliar application. Systemic pesticides may accumulate in the ginseng root following leaf spray [19]. However, buprofezin is not translocated [20]. Nevertheless, this pesticide was detected at all sites, albeit in different quantities. Ginseng stems thicken during long growth periods. As this enlarged stem is constantly disturbed by wind, it forms holes of various sizes in the areas where it directly contacts the soil. If these holes are enlarged, the rhizome may be exposed to the external environment, and the applied pesticide may flow through the stem and reach the roots. Pesticide sprayed onto the leaves might have reached the roots through the soil holes [21].

The residue levels at sites 1 and 3 did not significantly differ (*p* < 0.05). However, the residue levels at day 0 at Site 2 were ~2× higher than those at the other sites. The average residue level at harvest day, according to the Korean PHI, was 0.055 mg kg^−1^ at Site 2. However, the average residue levels at sites 1 and 3 were <LOQ. Table 1 shows that Site 2 received the lowest total amount of spray but had the highest residue level. Ginseng is usually grown by planting the seeds (2-y) and covering them with woven rice straw mats. Sites 1 and 3 were cultivated in this way, whereas Site 2 was cultivated without any rice straw mat. When cultivated ginseng is covered with rice straw mats, the rice straw absorbs the pesticide flowing into the roots, thereby reducing the pesticide residue levels [21]. Therefore, the average Site 2 residue levels were higher than those of the other sites.

### 3.4. Half-Lives and Prediction of the Residual Reduction Period

Time-dependent changes in buprofezin content in the ginseng at Site 2 were used to plot a regression curve. The reduction equation predicted the half-life and residual reduction period. Sites 1 and 3 could only be measured on days 0 and 7. The reduction equation obtained was y = 0.1638e^−0.0749x^ (Figure 3). The half-life of buprofezin in fresh ginseng was ~10 d. The residual buprofezin level decreased to <LOQ ~47 d after the final spraying. Buprofezin residues have been reported for numerous crops. Buprofezin half-lives are ~6 in pomegranates [20], 4.6 in eggplant [22], and 5.8–8.5 d in mango [23]. Hence, the buprofezin half-life may vary considerably among crops, possibly because of their relative differences in form and growth characteristics [24].

### 3.5. Processing and Reduction Factors

The residual pesticides were analyzed 14 d after the fresh ginseng was sprayed and processed into dried and red ginseng in accordance with the Korean PHI. In the dried ginseng at Site 1, the average residual buprofezin level was 0.046 ± 0.001 mg kg^−1^. At sites 2 and 3, the average residual buprofezin levels were 0.189 ± 0.002 and 0.013 ± 0.000 mg kg^−1^, respectively. In red ginseng, the buprofezin residue levels were 0.049 at Site 1, 0.171 at Site 2, and 0.006 mg kg^−1^ at Site 3. The residual buprofezin concentration increased after the fresh ginseng was processed into dried and red ginseng (Table 5).

If the processing factor was >1, the residual concentration increased during processing. If the reduction factor was <1, the absolute residual pesticide content decreased [17]. The processing and reduction factors could not be calculated for Sites 1 or 3, as the residual buprofezin concentrations were <LOQ. At Site 2, the mean processing factors for dried and red ginseng were 3.28 and 2.98, respectively (Table 6). Hence, their residual buprofezin concentrations were ~3× higher than those in raw (fresh) ginseng as the moisture content in the processed materials was ≤14%. However, the absolute buprofezin content in the dried and red ginseng decreased by ~20–30%, as the reduction factors calculated on the basis of their moisture content and yield were 0.81 and 0.71, respectively. Moreover, the buprofezin residue level was higher in the dried ginseng than the red ginseng.

The pesticide residue levels in agricultural products may change with processing. Dong (2012) [25] reported that in foods dried by sunlight and hot air, the residual pesticide levels may vary but are generally lower than those in the fresh materials. Compared with that in fresh grapes, the dimethoate level was 81% lower in sun-dried grapes, and 72% lower in hot air-dried grapes as dimethoate is thermolabile [9]. Additionally, methamidophos residue levels were reduced by evaporation during drying [10]. Furthermore, bitertanol residue levels were decreased by photolysis during sun drying but increased by hot air drying [11].

In contrast to the aforementioned reports, the present study showed that the relative buprofezin residue levels slightly decreased during ginseng processing. If buprofezin was thermolabile and photolabile and its vapor pressure and melting and boiling points were lower than the machining temperature, it should have been lost by decomposition and volatilization. However, buprofezin is both thermostable and photostable. Furthermore, its vapor pressure (4.2 × 10^−2^ mPa), melting point (−104–105.6 °C), and boiling point (267.6 °C) are all comparatively high [14]. Zhao et al. (2018) [12] reported that food processing concentrates pesticide residues and/or converts them into toxic metabolites, especially if the substances are thermostable and have high vapor pressures. In these cases, the residual concentrations in the dried materials should be higher than that in the fresh material.

## 4. Conclusions

The buprofezin residue levels in all ginseng samples harvested from three supervised field trials were below the Korean MRL. Thus, safe ginseng root can be produced when the ginseng leaves are sprayed in accordance with the Korean PHI. The residual buprofezin concentrations in dried and red ginseng were relatively higher than those in the fresh material. However, the absolute buprofezin content was slightly lower in the processed ginseng than in the fresh ginseng. As a rule, pesticide residues are strongly influenced by both food processing methods and pesticide physicochemical properties. However, buprofezin is a chemical with low vapor pressure, and high melting and boiling points; and even though heat is applied during ginseng processing, much of the pesticide residue is not lost. Although pesticides are necessary for effective crop production, their residue levels must be minimized in the final consumable product. Therefore, strategies are required to lower the pesticide residue levels in agricultural products during processing. In addition, the reduction factor should be reflected in the safety assessment as the concentration and the absolute amount of residual pesticides may vary depending on the processing method applied to other crops.

## Figures and Tables

**Figure 1 ijerph-18-00471-f001:**
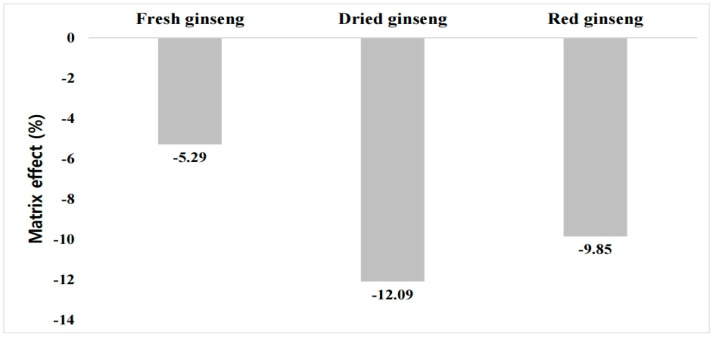
Buprofezin matrix effect for fresh ginseng and its processed products.

**Figure 2 ijerph-18-00471-f002:**
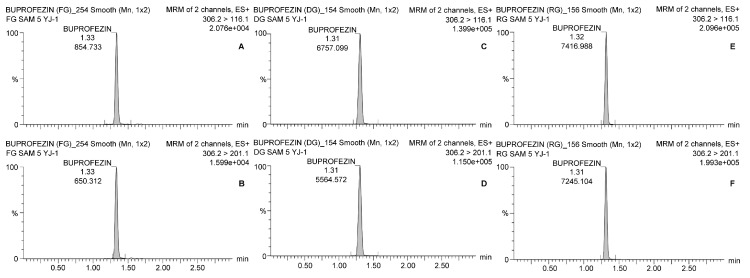
Chromatograms of buprofezin analysis with LC-MS/MS. The quantitation and confirmation ions were detected with 306.2 > 116.1 and 306.2 > 201.1, respectively. (**A**) Quantitation ion and (**B**) confirmation ion for fresh ginseng; (**C**) quantitation ion and (**D**) confirmation ion for dried ginseng; (**E**) quantitation ion and (**F**) confirmation ion for red ginseng.

**Figure 3 ijerph-18-00471-f003:**
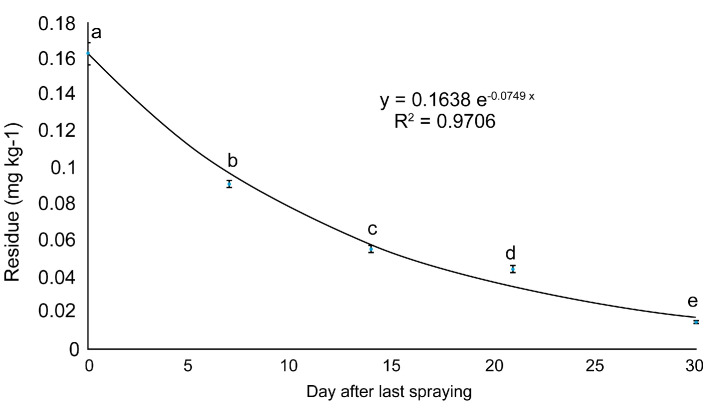
Dissipation curve of buprofezin residues in fresh ginseng at site 2. Values with the same letter in the same column are not significantly different at *p* < 0.05 according to the LSD test.

**Table 1 ijerph-18-00471-t001:** Buprofezin application doses and intervals on ginseng plants.

Site	Application No.	Application				
		Method	Volume (L ha^−1^)	Dose (kg a.i. ha^−1^)	Re-Treatment Interval (d)	Total Applied (kg a.i. ha^−1^)
I (Yeongju)	1	Foliar application	2016	0.252	-	0.774
	2	Foliar application	2070	0.259	10	
	3	Foliar application	2103	0.263	10	
II (Geumsan)	1	Foliar application	1646	0.206	-	0.608
	2	Foliar application	1580	0.198	10	
	3	Foliar application	1634	0.204	10	
III (Goesan)	1	Foliar application	1951	0.244	-	0.724
	2	Foliar application	1901	0.238	10	
	3	Foliar application	1938	0.242	10	

**Table 2 ijerph-18-00471-t002:** LC-MS/MS conditions for residual buprofezin analysis in fresh ginseng and its processing products.

<LC Condition>				
Instrument	Acquity UPLC H Class System, Waters (Milford, MA, USA)			
Column	Acquity UHPLC BEH C18			
Flow rate	0.3 mL min^−^^1^			
Mobile phase	0.1% (*v*/*v*) formic acid in water:0.1% (*v*/*v*) formic acid in acetonitrile (1:9, *v*/*v*)			
Injection volume	1 μL			
<Mass condition>				
Instrument	Triple-quadruple spectrometer, Xevo TQD, Waters (USA)			
Source temperature	150 °C		Cone gas flow rate	50 L h^−1^
Capillary voltage	3.0 kV		Ion source	ESI(+)
Desolvation gas	Temperature 200 °C; flow rate 650 L h^−1^		Scan type	MRM mode
<MRM condition>				
Precursor ion (*m/z*)	Quantitation ion		Confirmation ion	
	*m/z*	Collision energy	*m/z*	Collision energy
306.2	116.1	15	201.1	10

**Table 3 ijerph-18-00471-t003:** Statistical evaluation of buprofezin recovery in fresh ginseng and its processed products.

Ginseng	Fortification Level (mg kg^−1^)	Mean Value (%, *n* = 5)	Standard Deviation	Coefficient of Variation (%)
Fresh ginseng	0.005	91.09	1.87	2.05
	0.05	88.98	2.43	2.74
	0.07	88.86	2.43	2.73
Dried ginseng	0.005	83.85	3.15	3.75
	0.05	90.98	1.51	1.66
	0.07	89.24	1.27	1.42
Red ginseng	0.005	90.66	2.87	3.16
	0.05	96.04	1.83	1.91
	0.07	97.58	1.96	2.01

**Table 4 ijerph-18-00471-t004:** Residual buprofezin concentrations in fresh ginseng from three field trials.

Site	Mean Residue (mg kg^−1^, *n* = 3)				
	Days after Last Application				
	0	7	14	21	30
1	0.076 ^b^ ± 0.005	0.018 ^b^ ± 0.001	<LOQ	<LOQ	<LOQ
2	0.163 ^a^ ± 0.006	0.091 ^a^ ± 0.002	0.055 ± 0.002	0.044 ± 0.002	0.014 ± 0.002
3	0.078 ^b^ ± 0.002	0.012 ^b^ ± 0.000	<LOQ	<LOQ	<LOQ

Values with the same letter in the same column were not significantly different at *p* < 0.05 according to the LSD test.

**Table 5 ijerph-18-00471-t005:** Residual buprofezin concentrations in dried and red ginseng.

Site	Dried Ginseng			Red Ginseng		
	Residue (mg kg^−1^, *n* = 3)	SD	CV (%)	Residue (mg kg^−1^, *n* = 3)	SD	CV (%)
1	0.046 ^b^	0.001	1.87	0.049 ^b^	0.002	4.69
2	0.189 ^a^	0.002	0.81	0.171 ^a^	0.003	1.74
3	0.013 ^c^	0.000	2.08	0.006 ^c^	0.000	0.90

Values with the same letter in the same column are not significantly different at *p* < 0.05 according to the LSD test.

**Table 6 ijerph-18-00471-t006:** Buprofezin processing and reduction factor calculations for Site 2.

Rep.	Residue (mg kg^−1^)			Water Content (%)			Yield (%)		Residue Based on Dry Weight (mg kg^−1^)			Processing Factor		Reduction Factor	
	F	D	R	F	D	R	D	R	F	D	R	D	R	D	R
1	0.053	0.190	0.175	70.1	13.2	13.7	27.8	30.5	0.177	0.158	0.141	3.58	3.30	0.89	0.80
2	0.053	0.190	0.169	70.5	12.9	13.0	28.1	31.8	0.180	0.157	0.132	3.58	3.19	0.87	0.74
3	0.057	0.187	0.170	71.1	13.8	13.5	29.0	30.0	0.197	0.154	0.138	3.28	2.98	0.78	0.70
Mean	0.054	0.189	0.171	70.6	13.3	13.4	28.3	30.8	0.185	0.156	0.137	3.48	3.16	0.85	0.74

F, fresh ginseng; D, dried ginseng; R, red ginseng.

## Data Availability

Not applicable.

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
