# Peer review of "Effect of Processing on Residual Buprofezin Levels in Ginseng Products"

_ijerph, 2021, doi:10.3390/ijerph18020471_

Round 1

Reviewer 1 Report

This research manuscript highlighted the impact of ginseng processing conditions on buprofezin level. The manuscript is presented well with detailed extraction, purification and LC-MS/MS analysis. I have following queries and suggestion to the authors:

  • Does the title convey the meaning as “processing of buprofezin in the dried ginseng”? Please consider changing the title.
  • Give a brief on the effect of the processing on pesticide content in the introduction section.
  • I couldn’t find any citations claiming buprofezin in ginseng other than buprofezin in pomegranate (Utture et al. 2012), vegetables (Valverde-Garcia et al. 1998), mango (Mohapatra et al. 2020). If this is the first work, authors can mention it in the objective section.

Author Response

Answer to reviewer 1

Comments and suggestion for author

#1.     This research manuscript highlighted the impact of ginseng processing conditions on buprofezin level. The manuscript is presented well with detailed extraction, purification and LC-MS/MS analysis. I have following queries and suggestion to the authors:

Answer: Thank you very much for reviewing our manuscript and helping us improve its quality. Also, additional English editing was completed.

#2.     Does the title convey the meaning as “processing of buprofezin in the dried ginseng”? Please consider changing the title.

Answer: As you suggested, we revised the title to “Effect of processing on the buprofezin residues of dried ginseng products.”

#3.     Give a brief on the effect of the processing on pesticide content in the introduction section.

Answer: We have included relevant information regarding this in lines 42–50.

#4.     I couldn’t find any citations claiming buprofezin in ginseng other than buprofezin in pomegranate (Utture et al. 2012), vegetables (Valverde-Garcia et al. 1998), mango (Mohapatra et al. 2020). If this is the first work, authors can mention it in the objective section.

Answer: Related details are presented in lines 63–68, and we have added the reference for this information.

Reviewer 2 Report

The manuscript entitled “Buprofezin processing and reduction factors in dried ginseng products” presents the residual levels of the pesticide buprofezin in fresh ginseng and their change after processing. The subject of the paper is interesting since it concerns pesticides’ residues in food which is related to food quality and human health. However, the paper presents significant deficiencies which are reported below:

- Authors should highlight what is new in their work in relation to the existing literature.

- Authors should carefully revise the cited references to be in agreement with reported text.

- The conclusions are not presented in a clear way. Which is the factor that makes the pesticide concentration decrease and increase, in relation to the physicochemical properties of Buprofezin?

In more detail:

  • The abbreviation GAP in abstract should be explained.

Introduction

  • The introduction should be revised to include sufficient background on the occurrence of buprofezin in raw material like ginseng and processed food. In addition, it should be explicitly explained what is the “reduction factor” and include sufficient background on the existing methodology for its determination.
  • It is not understood how is reference [2] related to the text in Lines 33-34. The authors should revise and cite the appropriate literature.
  • It is not understood how is reference [7] related to the text in Lines 43-44. The authors should revise and cite the appropriate literature. Which pesticides have been found to be essential for ginseng production?
  • Line 46: Which pesticides may accumulate in ginseng?
  • Lines 46-47: “Moreover, pesticides applied to the soil before planting can be transferred to the ginseng”. Is this an hypothesis or a conclusion? The relevant literature reference should be added.
  • Lines 53-58: The authors should cite the appropriate reference for the reported information on buprofezin.

2.3 Reagents and materials

-The CAS No, the molecular and/or structural formula of the analyte buprofezin should be added.

2.4 Extraction and purification

- Line 111: The syringe filters that were used before LC-MS/MS analysis should be reported (material, and size) in Section Reagents and materials?

2.5 Optimization of instrumental analysis

- Line 113: Buprofezin has three nitrogen atoms and not two. The authors should correct it.

2.6 Matrix effect

- The equation (1) is not correct and not reported in the cited reference [11]. What does it mean “slope of matrix” and what does it mean “slope of matched standard calibration”?. Authors should explain and revise the equation. Which were the concentration levels that the matrix effect was investigated?

- Was a “blank sample” used for the validation of the method?

- Did the authors try to address the matrix effect by using a gradient elution program in order to improve resolution between the eluted matrix and analyte

- A chromatogram of a real sample should be presented in the paper.

2.7 Limits of quantitation and recovery

- The information is inconsistent. At which concentration levels were the recovery experiments applied? At 0.05 mg kg−1 and 0.25 mg kg−1 as reported in lines 139-140, or at 5 µg kg−1, 50 µg kg−1, and 70 µg kg−1 as reported in lines 143-144?

2.8 Processing and reduction factors

- Nor the equation (2), nor the term "reduction factor" is reported in the cited reference [12]. Authors should carefully revise and correct the reference.

- Equations (3) and (5) are the same. Remove appropriately. 

- Have the equations (2) to (7) been used in other relevant work?

3.1 Calibration curve and matrix effects

- What was the calibration range in mg/kg and how many were the calibration levels? What was the correlation coefficient of the linear regression lines?

- Figure 1 does not show any calibration curve. The text of lines 171-172 should be corrected. In addition Figure 1 should be revised to be more legible ex:” Matrix effect should be the title of the vertical axis and not of the horizontal.

3.2 Recovery

- Was the second MS/MS transition validated?

3.3 Residues in fresh ginseng

- The values in Table 4 are not consistent with the values in the text.

- How was half life calculated?

- The second sentence in the legend of Figure 2 seems irrelevant with the Figure.

- Did the authors investigate whether possible toxic metabolites of buprofezin were formed during processing according to the cited literature [25]?

- Table 6 is not legible. It should be improved.

Author Response

Answer to reviewer 2

Comments and suggestion for author

The manuscript entitled “Buprofezin processing and reduction factors in dried ginseng products” presents the residual levels of the pesticide buprofezin in fresh ginseng and their change after processing. The subject of the paper is interesting since it concerns pesticides’ residues in food which is related to food quality and human health. However, the paper presents significant deficiencies which are reported below:

Answer: Thank you very much for reviewing our manuscript and helping us improve its quality. Also, additional English editing was completed.

- Authors should highlight what is new in their work in relation to the existing literature.

Answer: We have added this information in the introduction.

- Authors should carefully revise the cited references to be in agreement with reported text.

Answer: We have revised the references throughout the manuscript.

- The conclusions are not presented in a clear way. Which is the factor that makes the pesticide concentration decrease and increase, in relation to the physicochemical properties of Buprofezin?

Answer: The effect of the physicochemical properties on the residual buprofezin levels in processed ginseng is described in the manuscript.

In more detail:

The abbreviation GAP in abstract should be explained.

Answer: We have defined the abbreviation ‘GAP’ in the abstract.

Introduction

The introduction should be revised to include sufficient background on the occurrence of buprofezin in raw material like ginseng and processed food. In addition, it should be explicitly explained what is the “reduction factor” and include sufficient background on the existing methodology for its determination.

Answer: We added this information in the introduction (lines 42–50).

It is not understood how is reference [2] related to the text in Lines 33-34. The authors should revise and cite the appropriate literature.

Answer: The main ingredients and pharmacological properties of ginseng are presented in reference [2].

It is not understood how is reference [7] related to the text in Lines 43-44. The authors should revise and cite the appropriate literature. Which pesticides have been found to be essential for ginseng production?

Answer: The contents of lines 43-44 are mentioned in this reference (now [13]). Various pesticides such as mancozeb, azoxystrobin, and carbendazim are used for growing ginseng, including the test pesticide buprofezin.

Line 46: Which pesticides may accumulate in ginseng?

Answer: Typically, systemic pesticides can accumulate. We have added this in the manuscript (line 57-58).

Lines 46-47: “Moreover, pesticides applied to the soil before planting can be transferred to the ginseng”. Is this an hypothesis or a conclusion? The relevant literature reference should be added.

Answer: Reference [13] contains the relevant information. We have added it (line 55).

Lines 53-58: The authors should cite the appropriate reference for the reported information on buprofezin.

Answer: Reference [14] contains the relevant information (line 68).

2.3 Reagents and materials

-The CAS No, the molecular and/or structural formula of the analyte buprofezin should be added.

Answer: The IUPAC name is presented in the introduction (line 65), and the Cas No. has been added in section 2.3 (line 107).

2.4 Extraction and purification

- Line 111: The syringe filters that were used before LC-MS/MS analysis should be reported (material, and size) in Section Reagents and materials?

Answer: We have added the information on the syringe filters (lines 116–117).

2.5 Optimization of instrumental analysis

- Line 113: Buprofezin has three nitrogen atoms and not two. The authors should correct it.

Answer: Thank you for your correction.

2.6 Matrix effect

- The equation (1) is not correct and not reported in the cited reference [11]. What does it mean “slope of matrix” and what does it mean “slope of matched standard calibration”?. Authors should explain and revise the equation. Which were the concentration levels that the matrix effect was investigated?

Answer: We had not written the correct equation. We have modified this. The range of concentration to investigate matrix effect is presented in line

155.

- Was a “blank sample” used for the validation of the method?

- Did the authors try to address the matrix effect by using a gradient elution program in order to improve resolution between the eluted matrix and analyte

Answer: We did not use a blank sample. The ME was calculated using the slope of the calibration curve of the standard made using an untreated sample elution and solvent.

- A chromatogram of a real sample should be presented in the paper.

Answer: We have added the chromatograms (Figure 3).

2.7 Limits of quantitation and recovery

- The information is inconsistent. At which concentration levels were the recovery experiments applied? At 0.05 mg kg−1 and 0.25 mg kg−1 as reported in lines 139-140, or at 5 µg kg−1, 50 µg kg−1, and 70 µg kg−1 as reported in lines 143-144?

Answer: This was incorrectly written, we have corrected it. The second values are the correct ones.

2.8 Processing and reduction factors

- Nor the equation (2), nor the term "reduction factor" is reported in the cited reference [12]. Authors should carefully revise and correct the reference.

Answer: Equation 2 is the calculation of the processing factor and is given in this reference (now [17]).

- Equations (3) and (5) are the same. Remove appropriately.

Answer: We have removed equation (5).

- Have the equations (2) to (7) been used in other relevant work?

Answer: The equations presented are commonly used.

3.1 Calibration curve and matrix effects

- What was the calibration range in mg/kg and how many were the calibration levels? What was the correlation coefficient of the linear regression lines?

Answer: We have added the range for the calibration curve and linear regression lines in section 3.1.

- Figure 1 does not show any calibration curve. The text of lines 171-172 should be corrected. In addition Figure 1 should be revised to be more legible ex:” Matrix effect should be the title of the vertical axis and not of the horizontal.

Answer: Thank you for your attention to detail. The content has been modified.

3.2 Recovery

- Was the second MS/MS transition validated?

Answer: No validation was performed. The first ion was used for quantification and the second one for qualitative analysis.

3.3 Residues in fresh ginseng

- The values in Table 4 are not consistent with the values in the text.

Answer: We have modified the values in the table.

- How was half life calculated?

Answer: As shown in section 3.4, a regression curve was derived using the residuals by sampling date, and half-life was calculated using this.

- The second sentence in the legend of Figure 2 seems irrelevant with the Figure.

Answer: It is associated with a regression curve based on the residuals by sampling date to calculate the half-life.

- Did the authors investigate whether possible toxic metabolites of buprofezin were formed during processing according to the cited literature [25]?

Answer: We did not study the metabolites of buprofezin.

- Table 6 is not legible. It should be improved.

Answer: We have revised the table.

Reviewer 3 Report

Buprofezin processing and reduction factors in dried  ginseng products

Authors Hyun Ho Noh et al.

In this work the authors evaluated the residual buprofezin levels in fresh ginseng and evaluated their changes during processing.

I understant that this paper addressed an anlytical method to evaluate the determination of buprofezin but I don’t see the content is to be published in IJERPH. I understand this investigation should be published in a journal with local interest but it has not international interest.

In my opinion this paper is very basic to be published in an International journal by the content and the work writing. The description of content is trivial. Information included in Table 1 and Table 2 are elemental to be included in Tables. They had to be included in the text. Equations 2 to 7 are basic.
I think this paper is interesting for a local/regional journal but I don't see the information is at the level demanded for to be published in an international journal.

Author Response

Answer to reviewer 3

Comments and suggestion for author

#1.     In this work the authors evaluated the residual buprofezin levels in fresh ginseng and evaluated their changes during processing. I understand that this paper addressed an analytical method to evaluate the determination of buprofezin but I don’t see the content is to be published in IJERPH. I understand this investigation should be published in a journal with local interest but it has not international interest. In my opinion this paper is very basic to be published in an International journal by the content and the work writing. The description of content is trivial.

Answer: Thank you very much for reviewing our manuscript and helping us improve its quality.

#2.     Information included in Table 1 and Table 2 are elemental to be included in Tables. They had to be included in the text.

Answer: We agree with you. However, we considered that presenting the analysis conditions and pesticide application doses as a table rather than text would make it clearer for the reader.

#3.     Equations 2 to 7 are basic.

Answer: We presented these in the manuscript to provide an accurate basis for the calculation of the processing and reduction factors. We believe that this will allow people outside the field and initiates to easily understand the contents of our manuscript.

#4.     I think this paper is interesting for a local/regional journal but I don't see the information is at the level demanded for to be published in an international journal.

Answer: We are aware that ginseng is not a crop that many people around the world are interested in. However, Korea produces a large quantity of ginseng and its processed derivatives and exports them to various countries, including Europe. Although there is not much production worldwide, we believe this is a necessary study to ensure the safety of pesticides for people who consume ginseng, including Europeans and Asians.

Reviewer 4 Report

The manuscript entitled “Buprofezin processing and reduction factors in dried ginseng products” reported some interesting results. The study determined residual buprofezin levels in fresh ginseng and evaluated their changes during processing. However, there are some questions should be resolved before further consider.

Major comments:

1. The poor discussion of the results. Authors just show the great amount of results that they have achieved, but they did not use them to develop an interesting discussion which could supplement to earlier studies.

2. Errors in grammar and language editing. Authors are responsible for preparing their papers in correct English language. The manuscript requires substantial grammatical revisions in its present form and it should not be accepted for publication unless both the technical and grammatical revisions have been made successfully and the English has been polished.

Technical comments:

1.In line 76 Table 1, the authorshouldredraw the three-wire table to maintain font consistency and remove the redundant lines.

2.Lines 88-91, acetonitrile and formic acid are both obtained from Merck (Darmstadt, Germany), and can be mergedthese two sentences.

3.In the section of “2.4 Extraction and purification”, 10-g, 50-mL, 1-mL, 2-mL, please deletethe hyphen in the middle.

4.Lines 98-105, “10 mL (Ten milliliters) acetonitrile was added …… sesquihydrate were sequentially added to the tube” this sentence was repeated twice.

5.Lines 135-137 Equation (1), the content of equation is too long. It is suggested that the authorsuse acronyms instead of long sentences, and then explain the specific meaning of the abbreviations under the equation.

6.In the section of “2.7 Limits of quantitation and recovery”, “0.05 mg kg−1 (10LOQ) and 0.25 mg kg−1 (50LOQ)” and “5 μg kg−1 (LOQ), 50 μg kg−1 (10 LOQ)”, what do the LOQ with number in parentheses mean respectively?

7.In line 172, (Figure 1) do not put it behind the calibration curve formula y = 75,780.2223x + 37.3711. Figure 1 shows the matrix effect.

8.In line 235, “Figure 2. Dissipation curve of buprofezin residues in 14-d samples”, is Figure 2 the dissipation curve of site 2? If so, the number of days should be 30 days instead of 14.

9.References: Many of the references have been superceded and more modern ones are required.

Author Response

Answer to reviewer 4

Comments and suggestion for author

The manuscript entitled “Buprofezin processing and reduction factors in dried ginseng products” reported some interesting results. The study determined residual buprofezin levels in fresh ginseng and evaluated their changes during processing. However, there are some questions should be resolved before further consider.

Answer: Thank you very much for reviewing our manuscript and helping us improve its quality.

Major comments:

  1. The poor discussion of the results. Authors just show the great amount of results that they have achieved, but they did not use them to develop an interesting discussion which could supplement to earlier studies.

Answer: The aim of this study was to assess whether the concentration of residual pesticides increased while processing ginseng and whether the absolute amount decreased. We believe the interpretation of the results regarding this approach is properly written. However, we would greatly appreciate it if you could specify what can be improved so that we can address it and hopefully move this manuscript closer to publication in the IJERPH.

  1. Errors in grammar and language editing. Authors are responsible for preparing their papers in correct English language. The manuscript requires substantial grammatical revisions in its present form and it should not be accepted for publication unless both the technical and grammatical revisions have been made successfully and the English has been polished.

Answer: We have submitted our revised manuscript to a professional English editing service.

Technical comments:

1.In line 76 Table 1, the authorshouldredraw the three-wire table to maintain font consistency and remove the redundant lines.

Answer: The table was slightly altered during formatting. We have reformatted it.

2.Lines 88-91, acetonitrile and formic acid are both obtained from Merck (Darmstadt, Germany), and can be mergedthese two sentences.

Answer: We have merged the two sentences according to your suggestion.

3.In the section of “2.4 Extraction and purification”, 10-g, 50-mL, 1-mL, 2-mL, please deletethe hyphen in the middle.

Answer: We have removed all hyphens.

4.Lines 98-105, “10 mL (Ten milliliters) acetonitrile was added …… sesquihydrate were sequentially added to the tube” this sentence was repeated twice.

Answer: We have merged the repeated text.

5.Lines 135-137 Equation (1), the content of equation is too long. It is suggested that the authorsuse acronyms instead of long sentences, and then explain the specific meaning of the abbreviations under the equation.

Answer: We have abbreviated the text in the equation and explained its meaning below.

6.In the section of “2.7 Limits of quantitation and recovery”, “0.05 mg kg−1 (10LOQ) and 0.25 mg kg−1 (50LOQ)” and “5 μg kg−1 (LOQ), 50 μg kg−1 (10 LOQ)”, what do the LOQ with number in parentheses mean respectively?

Answer: LOQ is the limit of quantitation, and 10LOQ means 10 times the LOQ. To clarify this, we added a multiplication sign.

7.In line 172, (Figure 1) do not put it behind the calibration curve formula y = 75,780.2223x + 37.3711. Figure 1 shows the matrix effect.

Answer: Section 3.1 refers to the calibration curve and matrix effect. Therefore, we presented the matrix effect results there.

8.In line 235, “Figure 2. Dissipation curve of buprofezin residues in 14-d samples”, is Figure 2 the dissipation curve of site 2? If so, the number of days should be 30 days instead of 14.

Answer: This was our mistake. Thank you for pointing it out. We have corrected it.

9.References: Many of the references have been superceded and more modern ones are required.

Answer: We agree that many of our references are outdated. This is because there aren’t many up-to-date papers related to our findings. However, the few latest papers have been consulted, although they are not cited in the manuscript. Because we only took note of it and didn’t quote it.

Round 2

Reviewer 2 Report

Authors improved significantly their manuscript addressing satisfactorily the reviewer’s comments. Please find below two minor comments on the revised manuscript.

  1. The title was changed to “Effect of various processing steps on residual buprofezin levels in processed ginseng products”. It is proposed to be simplified to “Effect of processing on residual buprofezin levels in processed ginseng products”
  2. Figure 1 should be revised to be more legible: “Matrix effect (%)” should be the title of the vertical axis and not of the horizontal.

Author Response

Response to Reviewer 2

Comments and suggestion for author

The title was changed to “Effect of various processing steps on residual buprofezin levels in processed ginseng products”. It is proposed to be simplified to “Effect of processing on residual buprofezin levels in processed ginseng products”

Answer: We agree with the suggested title, and we have modified the title accordingly.

Figure 1 should be revised to be more legible: “Matrix effect (%)” should be the title of the vertical axis and not of the horizontal.

Answer: Thank you for the suggestion. We have modified Figure 1 accordingly.

Reviewer 3 Report

This paper was corrected by the authors. It is OK for publication.

Author Response

Comments and Suggestions for Authors

This paper was corrected by the authors. It is OK for publication.

Reviewer 4 Report

The manuscript can be accepted for publication according to the revisions carried out.

Author Response

Comments and Suggestions for Authors

The manuscript can be accepted for publication according to the revisions carried out.